# Gramicidin Lateral Distribution in Phospholipid Membranes: Fluorescence Phasor Plots and Statistical Mechanical Model

**DOI:** 10.3390/ijms19113690

**Published:** 2018-11-21

**Authors:** István P. Sugár, Alexander P. Bonanno, Parkson Lee-Gau Chong

**Affiliations:** 1Department of Neurology, Icahn School of Medicine at Mount Sinai, New York, NY 10029, USA; 2Department of Medical Genetics and Molecular Biochemistry, The Lewis Katz School of Medicine at Temple University, Philadelphia, PA 19140, USA; tuf20683@temple.edu(A.P.B.)

**Keywords:** membrane organization, statistical mechanics, fluorescence spectroscopy, peptide-lipid interactions, gramicidins, lipid bilayers

## Abstract

When using small mole fraction increments to study gramicidins in phospholipid membranes, we found that the phasor dots of intrinsic fluorescence of gramicidin D and gramicidin A in dimyristoyl-*sn*-glycero-3-phosphocholine (DMPC) unilamellar and multilamellar vesicles exhibit a biphasic change with peptide content at 0.143 gramicidin mole fraction. To understand this phenomenon, we developed a statistical mechanical model of gramicidin/DMPC mixtures. Our model assumes a sludge-like mixture of fluid phase and aggregates of rigid clusters. In the fluid phase, gramicidin monomers are randomly distributed. A rigid cluster is formed by a gramicidin dimer and DMPC molecules that are condensed to the dimer, following particular stoichiometries (critical gramicidin mole fractions, *X*_cr_ including 0.143). Rigid clusters form aggregates in which gramicidin dimers are regularly distributed, in some cases, even to superlattices. At *X*_cr_, the size of cluster aggregates and regular distributions reach a local maximum. Before a similar model was developed for cholesterol/DMPC mixtures (Sugar and Chong (2012) *J. Am. Chem. Soc. 134*, 1164–1171) and here the similarities and differences are discussed between these two models.

## 1. Introduction

Gramicidin is an antimicrobial polypeptide produced by the soil bacterium *Bacillus brevis*. The native gramicidin, designated as gramicidin D, has 15 alternating D- and L-amino acids with three different components: 80% gramicidin A, 5% gramicidin B, and 15% gramicidin C [1,2]. Gramicidin A (HCO-L-Val-D-Gly-L-Ala-D-Leu-L-Ala-D-Val-L-Val-D-Val-L-Trp-D-Leu-L-Trp-D-Leu-L-Trp-D-Leu-L-Trp-NHCH_2_CH_2_OH) contains four tryptophans whereas gramicidin B and C have the tryptophan at position 11 replaced by phenylalanine and tyrosine, respectively. The N-terminus of the gramicidin polypeptide chain contains a formyl moiety (-HCO) and the C-terminus is linked to aminoethanol (-NHCH_2_CH_2_OH). Gramicidin is very hydrophobic and able to spontaneously insert into lipid bilayers. The peptide may span half of the liposomal membrane bilayer or form a head-to-head dimer of single helices, spanning the whole membrane and serving as a channel (diameter ~4 Å) passing monovalent cations across the membrane [3,4,5,6]. Gramicidins in phospholipid bilayers have been used as a valuable model system for studying membrane insertion and lipid-protein interactions of membrane-spanning channels [7]. In the present study, we used this model system to explore how channel peptides are distributed laterally in membranes, an area that has drawn increasing attention in recent years [8,9].

In this study, we focus on gramicidins/dimyristoyl-*sn*-glycero-3-phosphocholine (DMPC) mixtures because they have been studied extensively. An atomic-resolution structure of gramicidin A, in the form of the β^6.3^-helix, in oriented DMPC liquid crystalline bilayers has been reported [3,4,5]. Differential scanning calorimetry (DSC) [10,11] showed that small quantities of gramicidins totally remove the pre-transition endotherm of DMPC and broaden the lipid main phase transition. The enthalpy of the main phase transition decreases linearly with increasing gramicidin content until 0.05 gramicidin mole fraction and levels off at about 0.16 gramicidin mole fraction. Below 0.16 gramicidin mole fraction in DMPC, several interesting studies are noticed. Polarized attenuated total reflection infrared and spin-label electron spin resonance revealed that at a peptide/lipid molar ratio of 1:10, three to four lipids per monomer are motionally restricted by interaction with gramicidin A, probably via the association with the four tryptophans, which are located at the membrane-water interface [11]. Nuclear magnetic resonance (NMR) studies performed in the range of 0–0.05 gramicidin mole fraction pointed out that in gramicidin/DMPC mixtures each gramicidin molecule was surrounded by approximately one layer of bound lipids [12]. Earlier molecular dynamics simulations also indicated the presence of a layer of bound DMPC molecules around a gramicidin molecule [13]. More recent molecular dynamics simulations by Kim et al. [14] revealed the radial distribution of DMPC molecules both around a monomer and a dimer of gramicidin A. They found that the bilayer thickness strongly depends on the lateral distance from the gramicidin dimer. The membrane hydrophobic thickness changes non-monotonically from a value of 28 Å (gel state), with an initial decrease in the thickness within 4–5 Å from the dimer then an increase within the next 2.0–2.5 Å, followed by a monotonic decrease until it levels off at the thickness of fluid DMPC bilayer (25 Å). In contrast, the DMPC bilayer thickness around the gramicidin monomer remains at 25 Å unchanged with lateral distance. Based on NMR [3] and flash photolysis studies [15], Harroun et al. concluded that, at the gramicidin/DMPC molar ratio of 1:10, virtually all the gramicidin molecules (close to 100%) are in dimeric form [16]. These studies imply that, in gramicidins/DMPC mixtures, the lipid molecules around the peptide form the liquid ordered phase and the lipids away from the peptide are in liquid disordered phase.

The above-described structural features of gramicidin/DMPC mixtures in several aspects remind us of those seen in the cholesterol/DMPC mixtures, namely to the mixture of liquid ordered and liquid disordered phases. In the liquid ordered phase, phospholipid molecules are condensed to the cholesterol molecules while in the liquid disordered phase the phospholipid molecules around each cholesterol are in fluid phase. The above-described similarities between cholesterol/DMPC and gramicidins/DMPC encouraged us to find other similarities between these two mixtures. Namely, in the case of cholesterol/phospholipid mixtures several critical cholesterol mole fractions were found where biphasic changes in membrane properties were observed [17,18,19].

Similar to the model of cholesterol/phospholipid mixtures [17], here we develop a statistical mechanical model of gramicidin/DMPC mixture that predicts the existence of many critical mole fractions. This model provides a molecular description of lateral distribution of lipids and gramicidins and characterizes the biphasic changes around the critical gramicidin mole fractions. Two of the predicted critical mole fractions were detectable in nanosecond time-resolved data such as fluorescence lifetime phasor plots, but not obvious from steady-state measurements such as a membrane probe’s generalized polarization (GP), which suggests that the lateral structures of gramicidins/DMPC mixtures are similar to, but not the same as those found in cholesterol/DMPC mixtures. Several structural features of the lateral organization in gramicidins/DMPC mixtures are proposed.

## 2. Results

### 2.1. Phasor Plots of Intrinsic Protein Fluorescence in Gramicidins/DMPC Mixtures

In this study, the mole fraction of gramicidins in DMPC liposomes was kept below the solubility limit, ~0.15–0.18, beyond which fiber-like material appeared. As such, the highest possible critical gramicidin mole fraction in liposomes for displaying membrane biphasic behaviors is 0.143, if the sterol superlattice model [17,18,19,20] is applied to gramicidin/DMPC mixtures. To test this possibility experimentally, we prepared one set of gramicidin D (gD)/DMPC multilamellar vesicles (MLVs) having gramicidin mole fraction varied from 0.139 to 0.147 with a 0.02 mole fraction increment. In this sample set, 0.143 is the only critical mole fraction predicted to show a biphasic change in membrane properties [17]. The phase delay and demodulation of the intrinsic gD fluorescence (coming from tryptophans and tyrosines) of this sample set were measured using multiple modulation frequencies, and the phasor dots at each modulation frequency were determined.

Figure 1 shows the phasor dots of intrinsic gD fluorescence obtained from gD/DMPC MLVs at 37 °C. At this temperature, DMPC would be in liquid-crystalline state. It can be seen from Figure 1 that, at a given modulation frequency, especially those at the high end (e.g., 200 and 143.9 MHz, inlet of Figure 1), the phasor point for 0.143 mole fraction of gramicidin D (blue upward triangles) stands out while the phasor points for the other gramicidin mole fractions (i.e., 0.139, 0.141, 0.145, and 0.147) are clumped to a different area, thus forming a biphasic change in phasor dot at 0.143 mole fraction of gD. Movement of the phasor point by ~0.01 on the G or S axis (see Materials and Methods) is a statistically significant change, according to [21]. The relative errors of S and G (ΔS and ΔG) were calculated (see Materials and Methods) and their values (the ranges are given in the figure legend) are small (<0.0015). This set of data supports the idea that gD/DMPC mixtures are similar to cholesterol/DMPC mixtures in membrane lateral organization, both being able to exhibit a biphasic change in membrane behavior at a critical mole fraction, where regular distributions are dominating [19,20].

If the biphasic change in phasor dot at 0.143 mole fraction shown in Figure 1 is due to formation of regular distributions, then this phenomenon should be observable over a reasonably wide range of temperatures at which the membrane is in liquid crystalline state, according to the physical principles underlying lipid regular distribution [22]. Indeed, Figure 2 shows that a similar biphasic change in phasor dot at 0.143 mole fraction appears in gD/DMPC MLVs at 45 °C.

A biphasic change in phasor dot at 0.143 mole fraction not only occurs in gramicidin D (gD)/DMPC MLVs (Figure 1 and Figure 2), but also in gramicidin A (gA)/DMPC MLVs (Figure 3) at 37 °C. Gramicidin D is a mixture of gramicidins A (80%), B (5%) and C (15%), as mentioned earlier. Thus, our data show (Figure 1, Figure 2 and Figure 3) that the biphasic change in membrane behavior at a critical mole fraction holds regardless of the gramicidin type. Figure 4 further shows that this type of biphasic change in phasor dot persists when gA/DMPC large unilamellar vesicles (LUVs) (particle size: ~251 nm; polydispersity index (PDI): 0.25), instead of multi-lamellar vesicles, were examined.

Below the detected critical mole fraction $X_{cr}^{12}=0.143$ (where the notation $X_{cr}^{12}$ is explained at Table 1), we tried to measure the biphasic change in phasor dots around two other theoretically predicted critical gramicidin mole fractions: $X_{cr}^{13}=0.133$ and $X_{cr}^{14}=0.125$, however, the measured phasor dots did not show a biphasic change at the critical mole fractions (Appendix A). (Note that the critical mole fractions, $X_{cr}^{M}$, that were used in our calculations at different *M* values are listed in Table 1. Also listed in Table 1 are the cooperativity energies of the aggregation of the rigid clusters *w* (defined after Equation (S3)) for each *M*. These cooperativity parameters were calculated from our model by using Equations (S3) and (S11)–(S13) and the model parameters listed in Appendix A at two values of the energy parameter $\varepsilon_{g}^{s}-\varepsilon_{u}^{s}$.)

Then we tried to determine if there exists a biphasic change in phasor dot in a sample set around the predicted critical mole fraction just above 0.143 (i.e., $X_{cr}^{11}=0.154$) (Figure 5). In this sample set, the gD mole fraction in DMPC was varied from 0.147 to 0.160 (Figure 5). Figure 5B shows the phasor dots measured at and below 0.154 gD mole fraction of this sample set. Here, like in the case of critical mole fraction 0.143, the phasor dot belonging to the critical mole fraction (0.154) is located at the far left. However, the phasor dots measured at and above 0.154 are very close to each other and the phasor dot at 0.154 is not at the far left (Figure 5A). This result suggests that the solubility limit of gramicidins in DMPC is reached at the critical gramicidin mole fraction 0.154. Therefore, only the first half (≤0.154) of the biphasic change near the critical mole fraction 0.154 is observable.

Virtually all of the phasor dots presented in Figure 1, Figure 2, Figure 3, Figure 4 and Figure 5 are off the universal circle. This indicates that the intrinsic fluorescence intensity decay of gD and gA in DMPC bilayers is not a single exponential decay [23], which is expected as there are multiple tryptophans in each gramicidin, as mentioned earlier.

### 2.2. Generalized Polarization of Laurdan Fluorescence in Gramicidin A/DMPC Mixtures

To this end, we have time-resolved data (Figure 1, Figure 2, Figure 3, Figure 4 and Figure 5) from gramicidin fluorescence to show that a biphasic change in membrane properties occurs in gramicidin/DMPC mixtures at a critical mole fraction predicted by the sterol superlattice model. It is then of interest to test on the same sample set if a biphasic change in membrane properties can be revealed when the fluorescence comes from the membrane lipid matrix, not from the peptide gramicidin. For this reason, we labeled gramicidin A/DMPC mixtures with the membrane probe 6-lauroyl-2-(dimethylamino)naphthalene (Laurdan) and determined Laurdan’s generalized polarization (GP). As shown in Figure 6, Laurdan’s GP in gramicidin A/DMPC MLVs do not show a biphasic change with gramicidin mole fraction. In sharp contrast, in sterol/DMPC mixtures, a clearly discernible Laurdan GP dip was almost always observed at each critical sterol mole fraction examined [24].

To gain a molecular understanding of the experimental observations described above, we have built a statistical mechanical model to describe the lateral organization of gramicidin/DMPC mixtures.

### 2.3. Model

#### 2.3.1. On the Condensing Effect of Gramicidin

Molecular dynamics (MD) simulations determined the radial distribution and cross-sectional area of DMPC molecules around a dimer of gramicidin A (see Figures 7B and S9, respectively, in [14]). These simulations pointed out the condensing effect of gramicidin A dimer on the surrounding DMPC, but gramicidin monomer did not show condensing effect (see Figure S14 in [14]).

In a layer of a bilayer, the number of DMPC molecules surrounding the gramicidin dimer within a radius *R* can be calculated from the following integral:(1)$$\;N\left(R\right)=\rho_{DMPC}\int_{R_{g}}^{R}\left[g\left(R\right)2R\pi \right]dR\;$$
where g(R) is the radial distribution of DMPC molecules from the MD simulations [14]. $\rho_{DMPC}=\left(\frac{1}{62}\right)$ Å^−2^ is the lateral density of DMPC far from the gramicidin dimer (62 Å^2^ is the cross-sectional area of a DMPC in a one component fluid bilayer [25]) and $R_{g}=7.5$*Å* [14] is the radius of the gramicidin dimer.

Note that Equation (1) assumes centrosymmetric distribution of phospholipid molecules around the gramicidin molecule, an assumption that is not so good at small *N*. Let us imagine one layer of a bilayer that is built from similar units each containing one gramicidin and *N(R)* DMPC molecules. Figure 7 (red curve) shows the inverse of *2·N(R)* function. 

#### 2.3.2. Modeling Gramicidin/Phospholipid Mixture—Qualitative Description

We intend to model gramicidin/phospholipid mixtures close to the critical mole fractions. Similar to the model of cholesterol/phospholipid mixtures [17], we assume that at each critical mole fraction a densely and a loosely packed phase coexist in the bilayer. In general, an inhomogeneous system that is in thermal equilibrium with its surrounding is in a configuration that minimizes its free energy. This configuration is an optimal balance between low energy/low entropy and high energy/high entropy phases.

In our model, the high energy/high entropy phase contains gramicidin monomers and DMPC molecules where the phospholipid molecules are in fluid state and both gramicidin monomers and phospholipid molecules are able to diffuse laterally, similar to liquid disordered phase in cholesterol/DMPC mixtures [26]. The low energy/low entropy phase is represented by relatively rigid clusters, similar to condensed complexes in cholesterol/DMPC mixture [27]. At critical mole fraction $X_{cr}^{M}$, each rigid cluster is formed by $M=2\left[1\;-\;X_{cr}^{M}\right]/X_{cr}^{M}$ phospholipid molecules that are condensed to a central gramicidin dimer (cf. Equation (3) in [17]). Namely, *M/2* of the condensed lipids are in the upper and the other *M/2* condensed lipids are located in the lower layer of the bilayer. (In Table 1 critical mole fractions and the respective *M*’s are listed.) Within a rigid cluster of strongly interacting components, the lateral diffusion of the molecules is negligible. However, the rigid clusters are able to diffuse laterally in the loose phase and tend to aggregate with rigid clusters of similar size. Within the aggregate of rigid clusters, the gramicidin dimers are regularly distributed and a ‘superlattice’ of gramicidin dimers may be formed.

We develop a lattice model of the gramicidin/DMPC two-component bilayer. The bilayer is represented by a lattice where each lattice unit is a square (Figure 8). The surface area belonging to a lattice unit, $A_{M}$ is equal to the cross section of a rigid cluster (where M lipid molecules are condensed to the gramicidin dimer). In Figure 8, a rigid cluster (represented by a green square with a black dot at the middle) formed by a gramicidin dimer and *M* hydrocarbon chains condensed to the gramicidin dimer in the upper layer of the bilayer and another *M* hydrocarbon chains condensed to the gramicidin dimer in the lower layer of the bilayer. White squares in Figure 8 represent fluid lattice units where black and red circles are gramicidin monomers located in the lower and upper layer of the bilayer, respectively. In Figure 8A each rigid cluster contains 12 DMPC molecules condensed to a gramicidin dimer. A fluid lattice unit with two gramicidin monomers contains $11.1\left(=\frac{12\cdot 57.23}{62}\right)$ DMPC molecules. A fluid lattice unit with one gramicidin monomer contains $13.93\left(=11.1\;+\;\frac{7.5^{2}\cdot \pi }{62}\right)$ DMPC molecules. A fluid lattice unit with zero gramicidin monomer contains $16.8\left(=11.1\;+\;\frac{2\cdot 7.5^{2}\cdot \pi }{62}\right)$ DMPC molecules. Here the radius of a gramicidin is 7.5 Å [4,14]. The cross-sectional area of a DMPC, condensed to a gramicidin dimer, is 57.23 Å^2^
$=\left(A_{12}\;-\;7.5^{2}\cdot \pi \right)/6$ where the cross-sectional area of a rigid cluster at M = 12 is $A_{12}=520.1$ Å^2^ (see Figure 7). The cross-sectional area of a fluid DMPC is 62 Å^2^ [25]. Since 11.1 < 12, if each of the fluid lattice units would contain two gramicidin monomers in Figure 8A, $X_{g}$ would be larger than $X_{cr}^{12}$, i.e.: $X_{g}=0.1432≳X_{cr}^{12}=0.143$. Actually, in Figure 8A most of the fluid lattice units contain two gramicidin monomers but there are two fluid lattice units with one gramicidin monomer in each. Since in each of these two fluid lattice units (each containing one gramicidin monomer) the number of DMPC molecules is more than 12 thus $X_{g}=0.1427≲X_{cr}^{12}=0.143$. In Figure 8B most of the lattice units contain fluid DMPCs and zero gramicidin monomers and thus $X_{g}=0.0077\ll X_{cr}^{12}=0.143$. (In Figure 8B there are 25 gramicidin dimers and 1 gramicidin monomer, i.e., the total number of gramicidins is 51. Since $X_{cr}^{12}=0.143$ is the lower limit of critical mole fraction, at *X_g_* < 0.143, no more than 12 DMPC can condense to each gramicidin dimer and we assume that, at *X_g_* < 0.143, 12 DMPC molecules are condensed to each gramicidin dimer. In the 25 rigid clusters there are 25 × 12 = 300 DMPCs. At the lattice unit containing the gramicidin monomer there are 13.93 fluid DMPCs. In the remaining 400 − 25 − 1 = 374 fluid lattice units there are 374 × 16.78 = 6275.7 fluid DMPCs. Thus the gramicidin mole fraction is $X_{g}$ = (51/(51 + 300 + 13.93 + 6275.7)) = 0.0077.)

In Figure 8A,B, 97% and 98% of the gramicidins are in dimeric form, respectively (In Figure 8A there are 400 − 14 gramicidin dimers and 26 gramicidin monomers. Thus the percentage of the gramicidin dimers is $\frac{2\left(400\;-\;14\right)}{2\left(400\;-\;14\right)\;+\;26}100=96.74\%$. In Figure 8B there are 25 gramicidin dimers and 1 gramicidin monomer. Thus the percentage of the gramicidin dimers is $\frac{25\cdot 2}{25\cdot 2\;+\;1}100=98.04\%$. In Figure 8A,B the proportion of the membrane surface covered by aggregates of rigid clusters (where the gramicidin dimers are regularly distributed) is A_reg_ = 0.965 and 0.0625, respectively. (In Figure 8A out of 400 lattice units there are 400−14 rigid clusters, thus A_reg_ = (400−14)/400 = 0.965. In Figure 8B out of 400 lattice units there are 25 rigid clusters, thus A_reg_ = 25/400 = 0.0625) 

#### 2.3.3. Modeling Gramicidin/Phospholipid Mixture—Statistical Mechanical Description

Defining the Lattice

There are two types of lattice units: $N_{s}$ of them represent the rigid clusters (in Figure 8A these are green squares with black dot at their center), while the remaining $N_{u}$ lattice units represent the fluid phase of the system (in Figure 8A these are white squares with randomly distributed red and black circles). Note that in this lattice model rigid clusters are represented by squares, rather than circles, in order to get tight packing within the aggregates.

The following notations refer only to one layer of the bilayer: *n* is the number of gramicidin molecules (either monomer or part of a dimer), *m* is the number of phospholipid hydrocarbon chains, $N_{tot}=n+\left(\frac{m}{2}\right)$ is the total number of molecules, *M* is the number of hydrocarbon chains (in a layer of the bilayer) within a rigid cluster, $X_{g}=\frac{n}{N_{tot}}$ is the gramicidin mole fraction, $N_{s}$ is the number of rigid clusters (i.e., the number of gramicidin dimers), and $X_{g}^{s}=N_{s}/N_{tot}$ is the mole fraction of gramicidin molecules situated in rigid clusters.

##### Calculating $N_{u}$

$N_{u}$ is equal with the surface area of the fluid phase divided by the surface area of a lattice unit ($A_{M}$): (2)$$\;N_{u}=\frac{\left(n-N_{s}\right)A_{g}+\left(\frac{m-N_{s}M}{2}\right)A_{p}^{f}}{A_{M}}=N_{tot}\frac{\left(X_{g}-X_{g}^{s}\right)A_{g}+\left(1-X_{g}-X_{g}^{s}\frac{M}{2}\right)A_{p}^{f}}{A_{M}}\;$$
where $A_{g}$ and $A_{p}^{f}$ is the cross-sectional area of a gramicidin and the cross-sectional area of a DMPC in fluid phase (see Appendix A), respectively.

##### Free Energy of the Lattice

The half of the free energy of the lattice or the free energy of one layer of the bilayer is:(3)$$\;F=E_{u}+E_{s}+E_{i}-T\left(S_{u}+S_{s}+S_{u}^{mix}+S_{units}^{mix}\right)\;$$
where T is the absolute temperature, $E_{s}$ and $E_{u}$ are half of the internal energies of the s and u state lattice units, and $E_{i}$ is half of the interaction energy between the nearest neighbor lattice units. The half of the lattice entropy contains four terms: (1) and (2) half of the internal entropy of the *u* and *s* state lattice units $S_{u}$ and $S_{s}$, respectively, (3) the mixing entropy of the molecules of one layer of the bilayer within the fluid phase $S_{u}^{mix}$ and (4) half of the mixing entropy of the u and s state lattice units $S_{units}^{mix}$. These energy and entropy terms depend on the actual lattice configuration, (see Appendix A). 

##### On the Solubility Limit of Gramicidin

In our initial experiments we found that fiber-like material appeared beyond 0.18 mole fraction of gramicidin in gramicidin/DMPC mixtures and we estimated that the solubility limit of gramicidin is in the mole fraction region of 0.15–0.18. Later, based on the locations of the phasor dots measured around the predicted critical mole fraction ($X_{cr}^{11}=0.154$) we concluded that 0.154 is the solubility limit of gramicidin in DMPC bilayer (Figure 5). At this critical mole fraction to each gramicidin dimer 11 DMPC molecules are condensed, i.e., 5.5 at the upper and 5.5 at the lower level of the bilayer. This means that in a layer of the bilayer 6 DMPC are condensed to each gramicidin but one of the condensed DMPCs is shared with a laterally nearest neighbor gramicidin (Figure 9).

#### 2.3.4. Results of the Theoretical Model

The free energy function in Equation (3) can be used when the membrane is close to the critical gramicidin mole fractions, $X_{g}≅X_{cr}^{M}$ where the fundamental assumption of our model is most applicable, i.e., the system contains one type of rigid cluster each containing *M* phospholipid molecules condensed to one gramicidin dimer. At any given gramicidin mole fraction, $X_{g}$ which is close to a critical mole fraction $X_{cr}^{M}$, one can find the average number of the s state lattice units, $\langle N_{s}\rangle =N_{tot}\langle X_{g}^{s}\rangle$. For large $N_{tot}$ the position of the minimum of the free energy function (Equation (3)) well approximates the average of $X_{g}^{s}$ (see maximum term method in [28,29,30]). By using this average and Equation (2), one can calculate the average number of *u* state lattice units, $\langle N_{u}\rangle$. Finally, at given $X_{g}$ and *M*, the proportion of the area (of the gramicidin/phospholipid bilayer) covered by the *s* state lattice units is:(4)$$\;A_{reg}=\frac{\langle N_{s}\rangle }{\langle N_{s}\rangle +\langle N_{u}\rangle }\;$$

Within the area covered by *s* state lattice units the gramicidin dimers are regularly packed.

Figure 10 shows the calculated $A_{reg}$ vs. $X_{g}$ curves at several critical mole fractions. Note that the calculated $A_{reg}$ yields biphasic changes, showing local maximum at each critical gramicidin mole fractions, $X_{cr,}^{M}$ and A_reg_ only slightly depends on the value of the model parameter $\varepsilon_{g}^{s}-\varepsilon_{g}^{u}$ (see Appendix A).

Here it is important to mention that the model provides the same results (shown in Figure 10) at different coordination numbers, z as long as the values of the following products remain the same: $ze_{ss}$, $ze_{us}$ and $ze_{uu}$ (see Equation (S3)).

## 3. Discussion

### 3.1. On Measured and Predicted Critical Mole Fractions 

Our experiments showed a biphasic change in phasor dots of peptide intrinsic fluorescence at 0.143 gramicidin mole fraction in DMPC bilayers (Figure 1, Figure 2, Figure 3, Figure 4 and Appendix A). The biphasic change at this particular mole fraction is reproducible and does not occur by chance because it was observed with both gramicidin D and gramicidin A in both MLVs and LUVs at different temperatures on multiple independently prepared sample sets (see Appendix A for the results from two additional sample sets). The mole fraction 0.143 is one of the critical gramicidin concentrations predicted by our sludge-like superlattice model (see Table 1) and is the only predicted critical mole fraction in the mole fraction range of the sample sets examined (0.137–0.149). It is well known that intrinsic protein/peptide fluorescence lifetime is sensitive to protein/peptide conformational changes. Thus, the biphasic change in lifetime phasor in the neighborhood of 0.143 mole fraction (Figure 1, Figure 2, Figure 3 and Figure 4) suggests that gramicidin undergoes a conformational change when the gramicidin mole fraction is increased from slightly below 0.143 to 0.143. This conformational change is reversed when the mole fraction is slightly increased beyond 0.143. Further, the phasor dot at 0.143 mole fraction is always located at left compared to the neighboring non-critical mole fractions (Figure 1, Figure 2, Figure 3 and Figure 4). A shift of phasor dot to the left of the phasor plot indicates a longer fluorescence lifetime [31]. These results are consistent with our sludge-like superlattice model because the model predicts that the ordered superlattice area reaches a local maximum at the critical mole fractions (Figure 10) and because the fluorescence lifetime of Trp/Tyr is usually increased when the surrounding membrane packing is tighter and the water content is reduced.

### 3.2. On the Upper and Lower Limit of Critical Mole Fractions

A biphasic behavior of gramicidin fluorescence phasor dots is expected to also occur at other theoretically predicted critical mole fractions (Table 1). However, to this end, it is necessary to consider the applicability of our model and the possible technical difficulties for the observation of biphasic changes of membrane properties at other critical mole fractions. The first critical mole fraction, and consequently a maximum of A_reg_, is expected to be measured at the solubility limit of gramicidin in DMPC bilayers. According to our experiments (Figure 5) the critical mole fraction, $X_{cr}^{11}=0.154$, represents the solubility limit (Figure 10). Note that in this case the calculated maximum of A_reg_ at mole fraction 0.154 should be a half-maximum because the gramicidin starts to precipitate from the phosphatidylcholine lipid matrix above this critical mole fraction. Thus the actual solubility limit marks the upper limit of the applicability of our model. On the other hand, with increasing *M*, the predicted critical mole fractions become closer to each other and their reliable detection becomes increasingly difficult. Also, the condensing effect of the gramicidin should be weaker on phospholipid molecules that are farther away. Once the condensing energy at the perimeter of a rigid cluster is comparable with the thermal energy unit we reach the upper bound of the size of a rigid cluster, and the respective gramicidin mole fraction marks the lower limit of the applicability of our model. Since we could not detect a biphasic behavior of gramicidin fluorescence phasor dots at *M* > 12, the lower limit of critical mole fractions is $X_{cr}^{12}=0.143$. This could be the case because at *M* > 12 not only nearest neighbor but also next nearest neighbor DMPC molecules would belong to the rigid clusters and according to the molecular dynamics simulations [14] the next nearest neighbor DMPC molecules are in fluid state.

What happens if we decrease the gramicidin concentration from this lower limit of the critical mole fractions? According to Harroun et al. [16] even at *X* < 0.143 the majority of the gramicidins (98–100%) are still in dimeric form, and NMR studies performed in the range of 0–0.05 gramicidin mole fraction pointed out that in gramicidin/DMPC mixtures each gramicidin molecule was surrounded by approximately one layer of bound lipids [12]. Thus even at *X_g_* < 0.143 almost 100% of the gramicidins are part of rigid clusters and these rigid clusters may form aggregates. Since $X_{cr}^{12}=0.143$ is the lower limit of critical mole fraction at *X_g_* < 0.143 no more than 12 DMPC can condense to each gramicidin dimer and we assume that at *X_g_* < 0.143 12 DMPC molecules are condensed to each gramicidin dimer. As an example, a possible arrangement of the rigid clusters for *X_g_* = 0.0077 is shown in Figure 8B. With decreasing gramicidin concentration, the total area of the aggregates linearly decreases in the membrane, while the surface area of fluid DMPC linearly increases, i.e., A_reg_ linearly decreases from 1 to 0 as *X_g_* decreases from 0.143 to 0 (blue line in Figure 10, which shows only part of this linear decrease of A_reg_ from 1 to 0.98). The decrease is linear because the cross-sectional area of DMPC (located in the fluid phase) is the same no matter how close it is to the rigid aggregates.

### 3.3. Comparing the Results of the Model with Other Experimental Data

Beside the critical mole fractions at 0.143 and 0.154 predicted by our model and supported by our fluorescence lifetime measurements, other experimental results are also in agreement with our model’s prediction. At every critical mole fraction, the calculated *A*_reg_ is practically equal to 1 (Figure 10), i.e., at these mole fractions, there are only rigid clusters. Since in a rigid cluster the gramicidin is in dimeric form, there are only gramicidin dimers in the membrane at the critical mole fractions. Also, according to the model result, next to the critical mole fractions *A_reg_* is between 0.98 and 1. These results of our model, without any parameter fitting, are in agreement with the NMR [3] and flash photolysis studies [15], based on which Harroun et al. concluded that, at the 1:10 gramicidin/DMPC molar ratio, close to 100% of the gramicidin is in dimeric form [16].

### 3.4. Similarities and Differences between Gramicidin/DMPC and Cholesterol/DMPC Mixtures

Taken together, it appears from experimental data and theoretical calculations that gramicidins/DMPC and cholesterol/DMPC mixtures bear some similarities. The most remarkable similarities are: (i) a biphasic membrane dynamic behavior (Figure 1, Figure 2, Figure 3 and Figure 4) was measured and a biphasic change in the regular area fraction, A_reg_ (Figure 10) was calculated at a critical mole fraction 0.143. Similar biphasic changes were found at critical mole fractions of sterol/phosphatidylcholine (PC) mixtures (reviewed in [19]); and (ii) at the critical mole fractions, both the gramicidin near the Trp/Tyr residues (Figure 1, Figure 2, Figure 3, Figure 4, Figure 5 and Appendix A) and the sterol in DMPC [32] report a tighter membrane local environment.

However, there are significant differences between gramicidins/DMPC and cholesterol/DMPC mixtures. First, the biphasic membrane behavior at the critical gramicidin mole fraction 0.143 appears to be readily observable from time-resolved data such as tryptophan/tyrosine phasor dots (Figure 1, Figure 2, Figure 3 and Figure 4) [23], but not obvious from steady-state measurements such as Laurdan’s GP (Figure 6). In contrast, in sterol/PC mixtures, prominent Laurdan’s GP dips (a biphasic change in GP) were readily observable at critical mole fractions [24]. In addition, gramicidin does not increase Laurdan’s GP value in DMPC bilayers (comparing the data in Figure 6 with those in [33]) whereas cholesterol increases Laurdan’s GP in DMPC bilayers significantly, for example, from ~0.005 for 0 mole fraction cholesterol to ~0.3 for 0.15 mole fraction cholesterol at 37 °C [33]. The GP values depend on local polarity and dipolar relaxation [34], however, since GP values were calculated from steady-state fluorescence emission spectra, GP values alone do not permit the distinction between these two contributing factors.

Second, our model assumption came from the results of molecular dynamics simulations for cholesterol/DMPC [35] and gramicidin/DMPC [14]. Similar to the experimental differences between the lateral interactions of cholesterol/DMPC and gramicidin/DMPC, the molecular simulation results show fundamental differences too. In the case of cholesterol/DMPC simulations with increasing lateral distance from the cholesterol, the DMPC goes monotonically from gel to fluid state. In the case of gramicidin/DMPC, the change is not monotonic (see Figure 7B (red curve) in [14]). Nearest neighbor DMPCs to a gramicidin dimer are close to gel state, while the next nearest neighbors are closer to fluid state. Further increasing the distance from the gramicidin dimer the DMPC becomes again close to gel state and then with further increasing distance the DMPC monotonically approaches fluid state. Since DMPCs that are next nearest neighbors to a gramicidin dimer are in fluid state, they cannot condense to the gramicidin dimer. This explains that in gramicidin/DMPC mixtures the lower limit of the critical mole fraction is $X_{cr}^{12}=0.143$. At this critical mole fraction, only nearest neighbor DMPCs to a gramicidin dimer form the rigid cluster.

Third, for gramicidin/DMPC mixtures, the *A*_reg_ values are in the range of 0.98–1.0 (Figure 10), while in the case of cholesterol/DMPC mixtures, the *A*_reg_ values are in the range of 0.65–0.90 [17,18,19]. A possible reason of this difference is that the surface area of the side of a gramicidin is about 1.8–1.9 times larger than the surface area of the side of a cholesterol and that the gramicidin dimer, that is present only in the rigid clusters, spans the membrane, while cholesterol spans only one layer of the bilayer. Thus, in gramicidin/DMPC mixtures, the lipid molecules are able to condense to a larger surface. To calculate the surface areas, we used the following data: the length and radius of a gramicidin is 13.75 and 7.5 Å, respectively [3,14,36], while the length and radius of a cholesterol molecule is 16.3 and 3.29–3.48 Å, respectively [37,38].

These differences prompt us to propose that the condensing effect on DMPC by gramicidins is not the same as that by cholesterol. The condensing effect of cholesterol on neighboring lipid acyl chains is largely due to the rigid and flat alpha-face of the sterol tetracyclic ring [39]. It is conceivable that the tetracyclic ring of sterols undergoes little volume fluctuations over time due to structure constraints, thus the cholesterol’s condensing effect is less variant with time. In contrast, gramicidins do not have the same rigid and flat ring structure, and the segmental motions of gramicidins in lipid bilayers can occur especially when operating channel activities. Thus, compared to cholesterol, gramicidins in lipid bilayers are likely to have more structural fluctuations over time and consequently the condensing effect on lipid acyl chains by gramicidins may be more variant with time. It was reported that librational amplitudes of gramicidin A are as large as ±20° [40]. Due to the much lower Laurdan’s GP values in gramicidin/DMPC than in cholesterol/DMPC, it can be proposed that the lateral structures of gramicidins/DMPC mixtures are more fluid and more dynamic than those in cholesterol/DMPC mixtures. Perhaps, in gramicidins/DMPC mixtures, the rate of exchange between the rigid superlattice clusters and the fluid disordered domains is so high that only time-resolved techniques such as fluorescence lifetime phasor methods can readily detect the biphasic change in membrane structure in the neighborhood of critical mole fractions.

### 3.5. On Gramicidin/DMPE Mixtures

The results obtained from this study can be extended to other phospholipid membrane systems. However, it should be pointed out that condensation of phospholipid to gramicidin depends on the type of the polar headgroup of the lipid. For example, there was no decrease in transition enthalpy detected in gramicidin/1,2-dimiristoyl-*sn*-glycerol-3-phosphoethanolamine (DMPE) mixtures with increasing gramicidin ratio. It is assumed that instead of condensation of DMPE to gramicidin, the aggregation of gramicidin takes place in the bilayer [41]. This kind of component separation takes place when the average lateral interaction between similar components is stronger than the lateral interaction between different components [42].

### 3.6. Biophysical and Functional Implications

The mole fraction dependent behavior revealed in this study reflects how lipids and peptides are laterally organized in gramicidin/DMPC liposomal membranes. Since lateral organization is key to understanding membrane structure, this study paves the way to deepen our knowledge of the structure-activity relationship between gramicidins and phospholipids in model membranes. The results obtained from this study may also have significant functional implications. It has been proposed that, in addition to increased membrane permeability, the main cause of gramicidin’s killing of Gram positive bacteria is the induction of hydroxyl radicals through Fenton reaction and the subsequent depletion of nicotinamide adenine dinucleotide (NADH) from the tricarboxylic acid cycle [43]. Since the extent of sterol superlattice can affect free radical-induced sterol oxidation and surface acting enzyme activities [44,45,46], it is possible that the extent of gramicidin superlattice can affect gramicidin-induced hydroxyl radical production and bacteria cytotoxicity. In addition, the results from our current study may be applied to liposomes containing other antimicrobial peptides. Liposome encapsulation has been frequently used as a powerful strategy to protect and deliver antimicrobial peptides to target tissues for therapy [47]. 

## 4. Materials and Methods

### 4.1. Preparation of Gramicidin/DMPC Mixtures

Gramicidin A (Sigma, St. Louis, MO, USA) and gramicidin D (Behring Diagnostics, La Jolla, San Diego, CA, USA) were dissolved in either 2,2,2-trifluoroethanol (TFE) or methanol to create separate stock solutions. The concentrations of the gramicidin A and D stock solutions were determined by measuring absorbance at 280 nm in methanol on a Beckman DU-640 spectrometer (Fullerton, CA, USA) using the extinction coefficient 21,577 (our determination) and 20,700 M^−1^cm^−1^ [48], respectively. The stock solution of DMPC (Avanti Polar Lipids, Alabaster, AL, USA) was made in chloroform, with the phospholipid concentration determined as previously described [49]. Gramicidins and DMPC mixed in organic solvents were first dried under a stream of nitrogen gas and then subjected to high vacuum (1 × 10^−3^ mbar) for ~12 h using a Labcono freeze-dry system (Freezone 4.5, Kansas City, MO, USA). The dried DMPC/gramicidins were hydrated using a pre-warmed buffer (50 mM Tris, 10 mM EDTA, 0.02% NaN_3_, pH 7.15), followed by vortexing for 2–3 min at 40 °C to make multilamellar vesicles (MLVs) and annealing through three heating (40 °C for 0.5 h)/cooling (4 °C for 0.5 h) cycles. Unilamellar vesicles (LUVs) were made from MLVs by passing the vesicles 10 times through two stacked 400-nm Nucleopore polycarbonate membranes under nitrogen gas pressure at 40 °C using an extruder (Lipex Biomembranes, Vancouver, BC, Canada). The accuracy of the determination of gramicidin content in DMPC is estimated to be ~0.001–0.002 mole fraction, and the procedures to achieve such a high mole fraction accuracy have been described previously [50,51]. The particle size and polydispersity index (PDI) of the MLVs and LUVs were measured on a Malvern Nano ZS spectrometer (Worcs, UK).

### 4.2. Fluorescence Lifetime Measurements

Lifetimes of intrinsic gramicidin fluorescence in gramicidins/DMPC mixtures were determined on an ISS K-2 multifrequency phase-modulation fluorimeter using a 285-nm LED as the light source (ISS, Champaign, IL, USA). The emission was observed through a Hoya U360 filter. Phase and modulation values were determined relative to an ethanol solution of 2,5-diphenyloxazole (PPO, Aldrich, St. Louis, MO, USA; lifetime = 1.46 ns). For a given liposome sample, the phase and modulation values were measured as a function of modulation frequency three times. For each measurement at a given modulation frequency, hundreds of data acquisition iterations were employed until the maximum estimated standard error reached 0.2%. Typically, 15 modulation frequencies in the range 1–200 MHz were used. The phase (φ) and modulation (*M*) data are presented using phasor plots, which are a convenient and model independent way to describe complex fluorescence decays [23,52,53]. The phasor plots are constructed as S (=*M* sinφ) against G (=*M* cosφ) [23,54]. Each phasor dot was constructed using the average of S and G values determined from three independently prepared samples. The errors of S and G were calculated using the equations: $\Delta S=S\sqrt{{\left(s_{M}^{r}\right)}^{2}\;+\;{\left(\phi \cdot s_{\phi }^{r}\cdot ctg\left(\phi \right)\right)}^{2}}$ and $\Delta G=G\sqrt{{\left(s_{M}^{r}\right)}^{2}\;+\;{\left(\phi \cdot s_{\phi }^{r}\cdot tg\left(\phi \right)\right)}^{2}}$, where $s_{M}^{r}$ and $s_{\phi }^{r}$ are the relative error of M and φ, respectively.

### 4.3. Measurements of Generalized Polarization (GP) of Laurdan Fluorescence

Laurdan is a polarity sensitive fluorescent probe with its chromophore located near the diester phospholipid polar headgroup region. For fluorescence labeling, 1.1 µL of 1.055 mM Laurdan in dimethyl sulfoxide (DMSO) was added into 3 mL of liposome sample containing 450 nmol of lipid (probe-to-DMPC ratio ~1/400) at 37 °C and the samples were incubated at 37 °C for one hour prior to fluorescence measurements. The amount of DMSO added has previously been shown not to perturb membrane structure to any significant extent [55]. The emission spectra of Laurdan fluorescence were measured under gentle stirring at 37 °C on an ISS K2 fluorometer (Champaign, IL, USA). The background readings from vesicles without the probe were subtracted from the sample readings. The excitation generalized polarization (GP_ex_ = (I_435_ − I_500_)/(I_435_ + I_500_)) [56] was then calculated from the spectral readings. Here I_435_ and I_500_ are the fluorescence intensities at 435 nm and 500 nm, respectively. The sample was excited at 340 nm.

## 5. Concluding Remarks

The novelty of the current study is to reveal that the mole fraction of gramicidin relative to the matrix lipids is of fundamental importance for the behaviors of gramicidin/PC mixtures because membrane lateral structure can be altered significantly in a biphasic manner in the vicinity of a critical gramicidin mole fraction. In the long run, the finding of this kind may deepen our understanding of the structure-activity relationship between membrane lipids and membrane proteins/peptides. Specifically, in this study, we present a statistical mechanical model, which describes how gramicidin molecules are distributed in the plane of phospholipid, DMPC, membranes. This model is supported by our fluorescence phasor data and in good agreement with the data from NMR, flash photolysis and molecular dynamics simulations by others [3,14,36]. According to our model (namely, sludge-like gramicidin/DMPC superlattice model), at the critical mole fractions, the extent of the densely packed area reaches a local maximum, and the arrangement of gramicidin dimers in the densely packed area is ordered while in the region of fluid phase the arrangement of the gramicidin monomers is disordered. Close to the critical mole fractions in gramicidin/DMPC mixtures, the *A*_reg_ values are in the range of 0.98–1.0 (Figure 10), while in the case of cholesterol/DMPC mixtures, the *A*_reg_ values are in the range of 0.65–0.90 [17,19]. Otherwise, compared to sterols/PC mixtures, gramicidins/PC mixtures have the same thermodynamic tendency to form condensed complexes and subsequently ordered cluster aggregates and superlattices, in coexistence with the fluid phase.

## Figures and Tables

**Figure 1 ijms-19-03690-f001:**
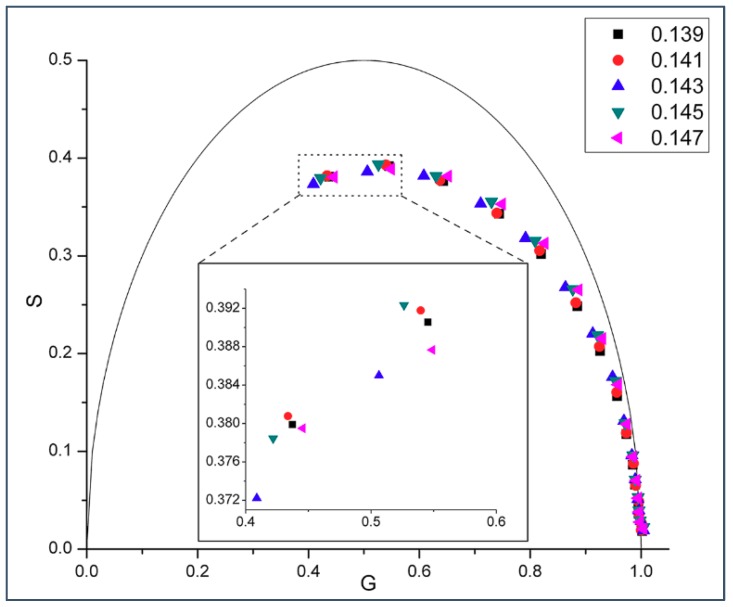
Phasor plot of intrinsic gramicidin fluorescence in various gramicidin D (gD)/DMPC multilamellar vesicles (MLVs) measured at 37 °C using 15 different modulation frequencies: (from left to right) 200.00, 143.94, 103.59, 74.55, 53.65, 38.61, 27.79, 20.00, 14.39, 10.36, 7.46, 5.37, 3.86, 2.78, and 2.00 MHz. The semi-circular arc is called the “universal circle” [21]. Inlet: enlarged phasor data measured at 200.00 and 143.94 MHz; the relative errors of G (=M cosφ) and S (=M sinφ) are: ΔG = 0.00098–0.00101 (200 MHz) and 0.00139–0.00141 (143.9 MHz), and ΔS = 0.00096–0.00099 (200 MHz) and 0.00096–0.00099 (143.9 MHz).

**Figure 2 ijms-19-03690-f002:**
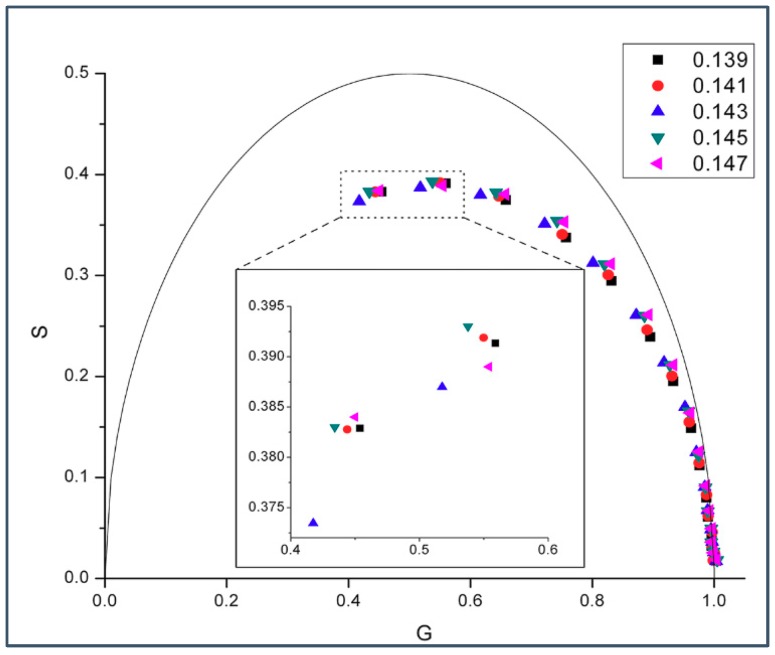
Phasor plot of gramicidin fluorescence lifetime in gD/DMPC MLVs with varying gD mole fractions ranging from 0.139–0.147. Samples were measured at 45 °C using 15 different modulation frequencies: (from left to right) 200.00, 143.94, 103.59, 74.55, 53.65, 38.61, 27.79, 20.00, 14.39, 10.36, 7.46, 5.37, 3.86, 2.78, and 2.00 MHz. The semi-circular arc is called the “universal circle” [21]. Inlet: enlarged phasor data measured at 200.00 and 143.94 MHz; the relative errors of G (=M cosφ) and S (=M sinφ) are: ΔG = 0.00097–0.001 (200 MHz) and 0.0013–0.00143 (143.9 MHz), and ΔS = 0.00097–0.00101 (200 MHz) and 0.00099–0.00101 (143.9 MHz).

**Figure 3 ijms-19-03690-f003:**
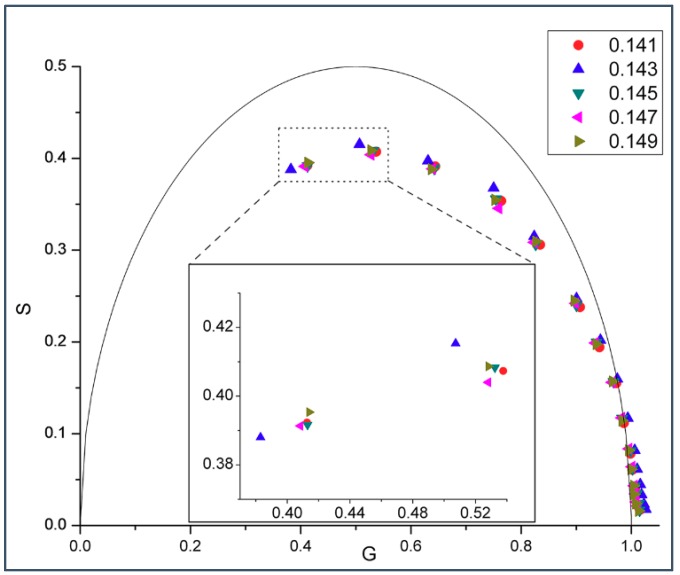
Phasor plot of gramicidin fluorescence lifetime in gramicidin A (gA)/DMPC MLVs. In this sample set, gA mole fraction was varied from 0.141 to 0.149 with an increment of 0.02. Samples were measured at 37 °C using 15 different modulation frequencies (from left to right) 200.00, 143.94, 103.59, 74.55, 53.65, 38.61, 27.79, 20.00, 14.39, 10.36, 7.46, 5.37, 3.86, 2.78, and 2.00 MHz. The semi-circular arc is called the “universal circle” [21]. Inlet: enlarged phasor data measured at 200.00 and 143.94 MHz; the relative errors of G (=M cosφ) and S (=M sinφ) are: ΔG =0.00101–0.00102 (200 MHz) and 0.00146–0.00150 (143.9 MHz), and ΔS = 0.00099–0.00101 (200 MHz) and 0.00101–0.00102 (143.9 MHz).

**Figure 4 ijms-19-03690-f004:**
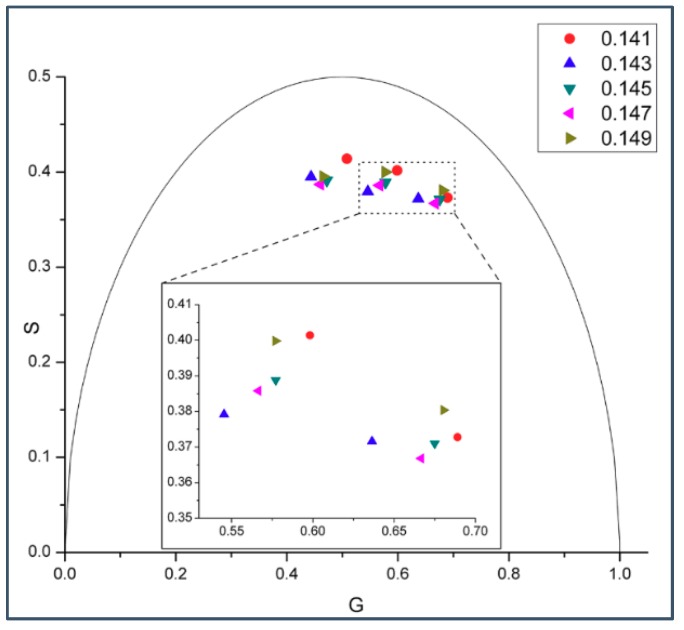
Effect of gA mole fraction on the phasor dots of gramicidin fluorescence lifetime in gA/DMPC large unilamellar vesicles (LUVs). In this sample set, five gA mole fractions (0.141, 0.143, 0.145, 0.147, 0.149) were examined. Samples were measured at 37 °C using 3 different modulation frequencies (from left to right) 200.00, 143.94, and 103.59 MHz. The semi-circular arc is called the “universal circle” [21]. Inlet: enlarged phasor data measured at 143.94 and 103.59 MHz; the relative errors of G and S are: ΔG = 0.00136–0.00145 (143.9 MHz) and 0.00184–0.00192 (103.59 MHz), and ΔS = 0.00097–0.00103 (143.9 MHz) and 0.00091–0.00095 (103.59 MHz).

**Figure 5 ijms-19-03690-f005:**
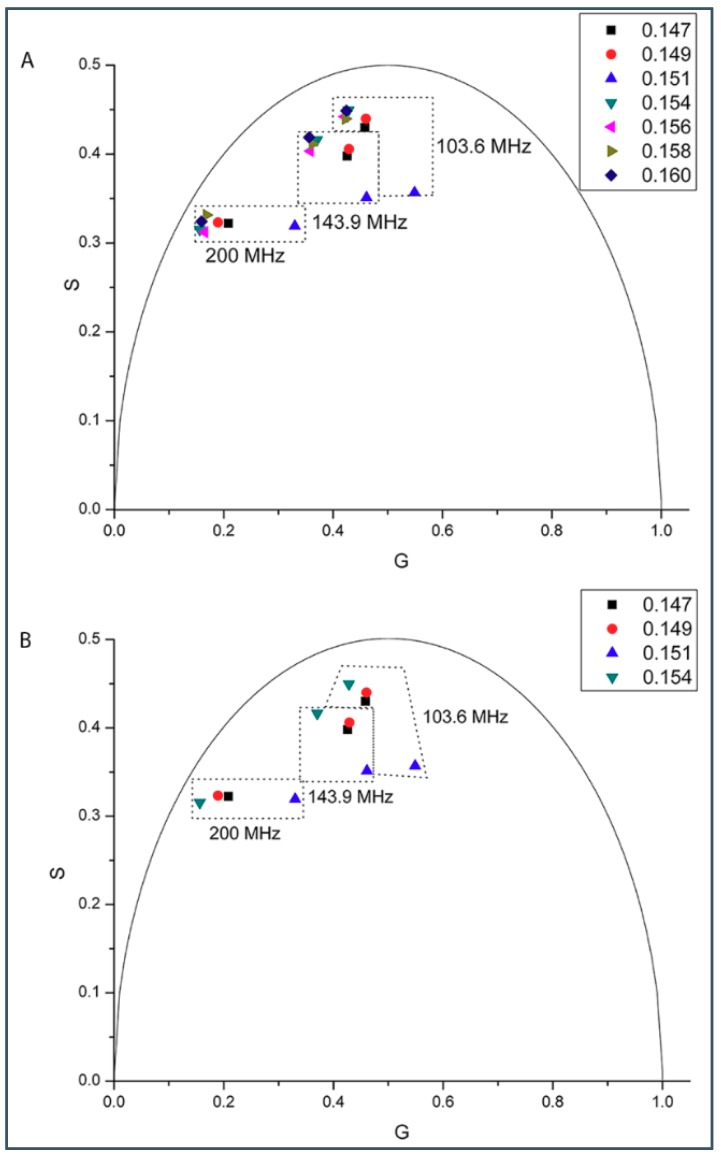
Phasor plot of gramicidin fluorescence lifetime in a sample set of gD/DMPC MLVs with gD content centered around the theoretically predicted critical mole fraction 0.154. Samples were measured at 37 °C using 3 different modulation frequencies (from left to right) 200.0, 143.9 and 103.6 MHz; the relative errors of G and S are: ΔG = 0.000824–0.015405 (200 MHz), 0.001269–0.002807 (143.9 MHz), and 0.002025–0.031970 (103.6 MHz) and ΔS = 0.000632–0.000815 (200 MHz), 0.000879–0.000949 (143.9 MHz), and 0.000849–0.000902 (103.6 MHz). The semi-circular arc is called the “universal circle” [21]. At every modulation frequency the phasor dots were measured at the following mole fractions: gD mole fraction: (**A**) 0.147, 0.149, 0.151, 0.154, 0.156, 0.158, 0.160; (**B**) 0.147, 0.149, 0.151, 0.154.

**Figure 6 ijms-19-03690-f006:**
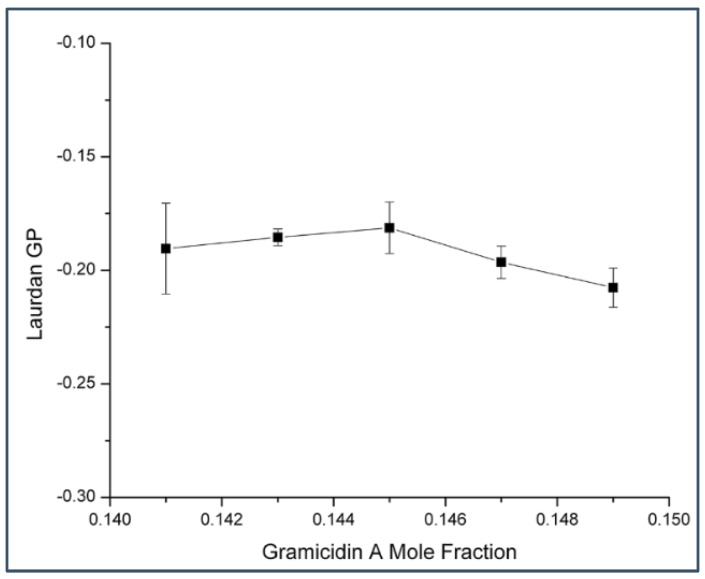
Laurdan’s generalized polarization (GP) versus gramicidin A mole fraction in gramicidin A/DMPC MLVs. Temperature = 37 °C. Error bars are the standard deviations of GP values obtained from three independently prepared samples.

**Figure 7 ijms-19-03690-f007:**
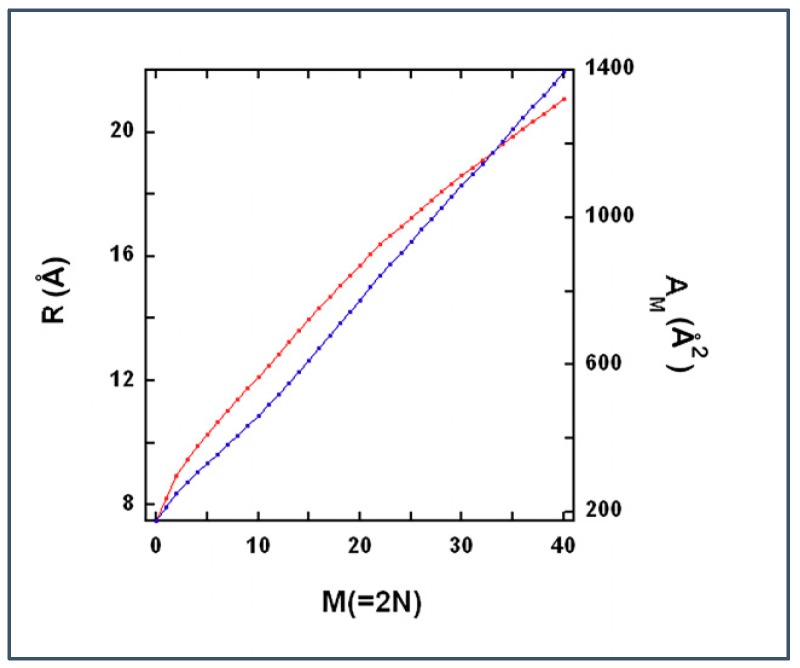
Condensing effect of gramicidin dimer. Red and blue curves are the radius, R and cross-sectional area, $A_{M}=R^{2}\pi$, respectively, of a rigid cluster as a function of the number of hydrocarbon chains, within a layer of the bilayer, condensed to a gramicidin dimer, M (=2 N). These curves were calculated from Equation (1) by using parameter values Rg = 7.5 Å [14]. R and $A_{M}$ are given in Å and Å^2^, respectively.

**Figure 8 ijms-19-03690-f008:**
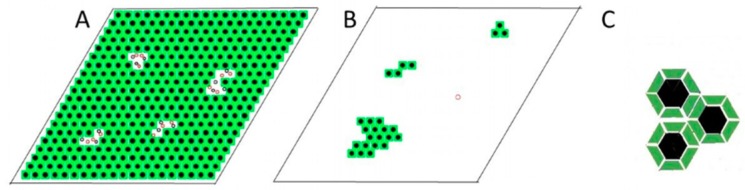
Lattice model of gramicidin/DMPC membrane. The bilayer is represented by hexagonally arranged units of squares. The surface area of a unit is equal with the surface area of a rigid cluster, $A_{M}$ (see Figure 7). A unit represents either a rigid cluster (green unit with black dot at the center) or part of the fluid phase (white unit with randomly distributed black and red circles). Black dot: gramicidin dimer. Green square: phospholipid molecules condensed to the central gramicidin dimer. Red and black circle: gramicidin monomer in the upper and lower layer of the bilayer, respectively. (**A**) $X_{g}=0.1427\approx X_{cr}^{M}=0.143$; (**B**) $X_{g}=0.0077\ll X_{cr}^{M}=0.143\;$ where M = 12; (**C**) Gramicidin dimers (black hexagons) may be regularly distributed into superlattices in the aggregated rigid clusters. This is an illustration of an aggregate of 3 rigid clusters where 12 phospholipids (green trapezoids) are condensed to each gramicidin dimer (6 located at the upper and 6 at the lower layer of the bilayer, i.e., this is the case when M = 12).

**Figure 9 ijms-19-03690-f009:**
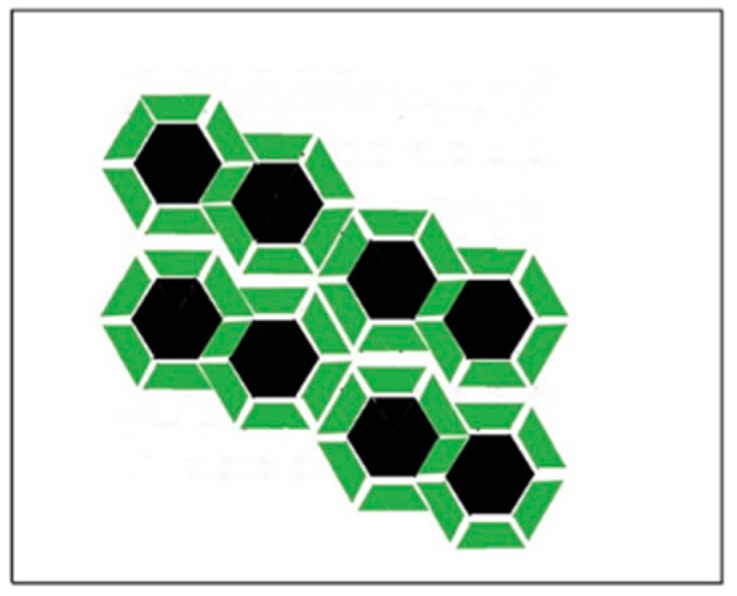
Aggregate of rigid clusters at $X_{g}≅X_{cr}^{M}=0.154$ where M = 11. Gramicidin dimers (black hexagons) may be regularly distributed into superlattices in the aggregated rigid clusters. This is an illustration of an aggregate of 8 rigid clusters where 11 phospholipids (green trapezoid: lipid condensed to one gramicidin dimer; green parallelogram: lipid condensed to two nearest neighbor gramicidin dimers) are condensed to each gramicidin dimer (5.5 located at the upper and 5.5 at the lower layer of the bilayer).

**Figure 10 ijms-19-03690-f010:**
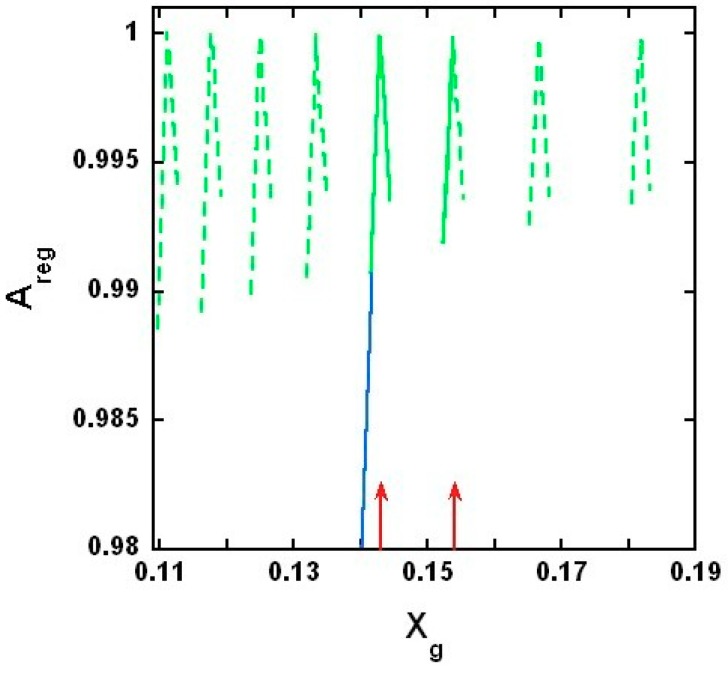
Proportion of regularly packed membrane area. Regular area fraction, $A_{reg}$ is plotted against the gramicidin mole fraction, $X_{g}$. Green lines: the curves of regular area fractions calculated around the critical gramicidin mole fractions, by Equation (4). Green dashed lines are theoretically calculated but not experimentally supported. At $X_{g}<0.143$ the theoretically predicted peaks would appear experimentally if more than 12 DMPC’s were able to condense to a gramicidin dimer. Green solid lines are theoretically calculated and experimentally supported. One of the red arrows is at the measured lower limit of critical mole fractions, (Figure 1, Figure 2, Figure 3 and Figure 4). The other red arrow is at the measured upper limit of the critical mole fractions, (Figure 5) which is also the solubility limit of gramicidin in DMPC bilayer. Gramicidin precipitates from the gramicidin/DMPC bilayer above this mole fraction (see explanation to Figure 5). The blue line is the assumed change of $A_{reg}$ if there is no critical gramicidin mole fraction below 0.143. If the vertical axis started from $A_{reg}=0$ and the horizontal axis from $X_{g}=0$, then the blue line would go from ($A_{reg}=1$, $X_{g}=0.143$ ) to ($A_{reg}=0$, $X_{g}=0$ ). The model parameters are listed in Appendix A and the energy difference was: $\varepsilon_{g}^{s}-\varepsilon_{g}^{u}=0$ cal/mol.

**Table 1 ijms-19-03690-t001:** Cluster Characteristics at Coordination Number z = 4.

*M*	$\;X_{cr}^{M}\;$	*w* (cal/mol)	*w* (cal/mol)
		$\;\varepsilon_{g}^{s}-\varepsilon_{u}^{s}=0\;$	$\;\varepsilon_{g}^{s}-\varepsilon_{u}^{s}=-1000\;$
9	0.182	489.2	447.0
10	0.166	486.8	444.1
11	0.154	485.2	442.3
12	0.143	484.1	441.5
13	0.133	483.7	441.4
14	0.125	483.8	442.0
15	0.118	484.3	443.2
16	0.111	485.3	444.9
17	0.105	486.6	447.0
18	0.100	488.3	450.0
19	0.0952	490.2	452.4
20	0.0909	492.4	455.5
21	0.0870	494.8	459.0
22	0.0833	497.4	462.5
23	0.0800	500.3	466.4
24	0.0769	503.3	470.4
25	0.0741	506.4	474.5

* Note, that in the case of coordination number z = 6 the cooperativity energies listed in Table 1 should be multiplied by 2/3. The critical mole fractions, $X_{cr}^{M}$, are calculated from the number of DMPC condensed to the gramicidin dimer, *M*, as follows: $X_{cr}^{M}=\frac{1}{1+\left(\frac{M}{2}\right)}$.

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
