# Peer review of "Gramicidin Lateral Distribution in Phospholipid Membranes: Fluorescence Phasor Plots and Statistical Mechanical Model"

_ijms, 2018, doi:10.3390/ijms19113690_

Reviewer 1 Report

Dear Authors,

It is a nice manuscript composed of experimental and computation work. However the computation part may be longer than it should be. “2.3 Model takes about half of the text (without the References).  In this chapter the size of the four page long statistical mechanical description (2.3.3) may also be reduce. Kindly try to shrink the computation part of the manuscript.

Reviewer 2 Report

The authors study the gramicidin lateral distribution in DMPC Membranes using fluorescence phasor plots and statistical mechanical model. The authors observed a similar behavior of gramicidin/DMPC to that observed previously in cholesterol/DMPC mixtures.

The main drawback of the manuscript is the excessive comparation between gramicidin behavior and cholesterol at the membrane level. Gramicidin is chemically very different from cholesterol and the comparisons regarding the rigid clusters that both form at the bilayer level should not be exposed in the abstract of the paper.

Furthermore, gramicidin dimeric behavior is quite explored, and the novelty of this paper should be highlighted. The authors fail to address how the concentration dependent behavior is key in the understanding of gramicidin interaction with model membranes. Biophysical and clinical implications could be extended to link the theoretical analysis with its application. These aspects must be significantly improved before the work is ready for publication.

Major remarks:

1- In the abstract the authors say “This model is similar to the model of 21 cholesterol/DMPC mixtures (Sugar and Chong (2012) J. Am. Chem. Soc. 134, 1164-1171.). However, 22 there are differences too such as: 1) in the case of cholesterol/DMPC mixtures there are only 23 cholesterol monomers in both the fluid phase and the aggregates of rigid clusters, 2) in the case of 24 cholesterol/DMPC mixtures ten, while in the case of gramicidin/DMPC mixtures only two, critical 25 concentrations were detected, 3) at Xcr, the proportion of the area of rigid clusters Areg in 26 gramicidin/DMPC mixtures is in the range 0.98-1.0, which is much higher than that (0.65-0.90) for 27 cholesterol/DMPC mixtures.” The authors should take the reference to a previous study from the abstract and take this considerations to the discussion part.

2- In lines 99/100 authors say that they have prepared “multilamellar vesicles (MLVs) (particle size: ~1030 nm; PDI: 1.0)”. A PDI of 1.0 is absolutely polydisperse sample, and there is no way with such a high polydispersity to be able to measure the size of the MLVs, simply because you cannot have a good correlogram. Therefore, saying the size is 1030 does not mean anything as this size is for sure not correct.

3- In lines 175 the authors say “Virtually all of the phasor dots (Figures 1-5) are off the universal circle. This indicates that the intrinsic fluorescence intensity decay of gD and gA in DMPC bilayers is not a single exponential decay [23], which is expected as there are multiple tryptophans in each gramicidin, as mentioned earlier.”

Accordingly the authors should do phasor plots for a multi-exponential decay time instead of  a single exponential decay. That is, the phasors for multiexponential lifetime decays or combinations of single-exponential lifetime decays given by the normalized linear combination of the component phasors, making them fall below the universal circle.

4- Line 612 – The authors included gramicidin in the organic phase with DMPC, prepared MLVs by the thin film hydration method and afterwards prepared LUVs by extrusion. My concern is if the authors are sure that gramicidin was not retained by the filter pore. Despite a 400 nm pore, even the lipids (about 10%) are retained during extrusion. Gramicidin should be quantified previously and after extrusion.

 Minor corrections:

-There are several missing symbols throughout the manuscript. E.g. Line 13, 48 and 50 is missing an “alpha”; Lines 113, 122, 123, 134, 135, 148, 149, 155 is missing a “delta”. Also missing symbols on the legend of figure 7

- Legends of figure 2, 4, 5 and 6, the symbol of grade (°C) should be corrected

- Legend of figure 7, in Å2 the number “2” should be in superscript

- Figure 7, I suggest that in the axis the unit Angstron is expressed with a symbol and not by the word

- Quality of the figures is not good and should be improved

- Line 299 – the subtitle “calculating” is not appropriate and should be changed

- Line 160 - "Data not shown" should be avoided in research manuscripts. Authors should provide the reference indicating the “data not shown” and cite the reference or put the data as supplementary

- Authors should keep coherence. In the text, the molar fractions are referred as: 0.139, 0.141, 0.145, and 0.147, but in the figure legends the same molar fractions are presented in percentages 13.9%, 14.1%, 14.5%, and 14.7%. Only one of this format should be used.

 Author Response

Round  2

Reviewer 2 Report

The authors have answered to all my queries and improved the content of the manuscript.

There are still the same format issues that I pointed out and that persisted in the revised form, therefore I recommended ijms editorial to take notice of this problem.